# 3D Multicellular Stem-Like Human Breast Tumor Spheroids Enhance Tumorigenicity of Orthotopic Xenografts in Athymic Nude Rat Model

**DOI:** 10.3390/cancers13112784

**Published:** 2021-06-03

**Authors:** Reza Bayat Mokhtari, Bessi Qorri, Manpreet Sambi, Narges Baluch, Sushil Kumar, Bikul Das, Myron R. Szewczuk, Herman Yeger, Hai-Ling Margaret Cheng

**Affiliations:** 1Department of Biomedical and Molecular Sciences, Queen’s University, Kingston, ON K7L 3N6, Canada; rbm7@queensu.ca (R.B.M.); bessi.qorri@queensu.ca (B.Q.); 13ms84@queensu.ca (M.S.); 2Department of Experimental Therapeutics,Thoreau Laboratory for Global Health,M2D2, University of Massachusetts, Lowell, MA 01852, USA; bdas@kavikrishnalab.org; 3Department of Immunology and Allergy, The Hospital for Sick Children, Toronto, ON M5G 0A4, Canada; Narges.baluch@sickkids.ca; 4QPS, Holdings LLC, Pencader Corporate Center, 110 Executive Drive, Newark, DE 19702, USA; sushilkumar29@yahoo.ca; 5Department of Cancer and Stem Cell Biology, KaviKrishna Laboratory,GBP, Indian Institute of Technology, Guwahati 781039, India; 6Program in Developmental and Stem Cell Biology, The Hospital for Sick Children, Toronto, ON M5G 0A4, Canada; hermie@sickkids.ca; 7The Edward S. Rogers Sr. Department of Electrical & Computer Engi-neering, Institute of Biomedical Engineering, University of Toronto, Toronto, ON M5G 1M1, Canada; 8Translational Biology & Engineering Program, Ted Rogers Centre for Heart Research, University of Toronto, Toronto, ON M5G 1M1, Canada

**Keywords:** multicellular 3D spheroid culture, orthotopic tumor xenograft, cancer stem cells

## Abstract

**Simple Summary:**

Breast cancer presents a unique clinical problem because of the variety of cellular subtypes present, including cancer stem cells (CSCs). Breast CSCs can induce the formation of new blood vessels at the site of tumor growth and a develop metastatic phenotype by enhancing a stromal cell response, similar to that of the primary breast cancer. The aim of this study was to investigate breast cancer cells cultured in stromal stem cell factor-supplemented media to generate 3D spheroids that exhibit increased stem-like properties. These 3D stem-like spheroids reproducibly and efficiently established orthotopic breast cancer xenografts in the athymic nude rat. This approach enables a means to develop orthotopic tumors with a stem-like phenotype in a larger athymic rat rodent model of human breast cancer.

**Abstract:**

Therapeutic targeting of stem cells needs to be strategically developed to control tumor growth and prevent metastatic burden successfully. Breast cancer presents a unique clinical problem because of the variety of cellular subtypes present, including cancer stem cells (CSCs). The development of 3D stem-like properties of human breast tumor spheroids in stem cell factor conditioned media was investigated in orthotopic xenografts for enhanced tumorgenicity in the athymic nude rat model. MCF-7, ZR-75-1, and MDA-MB-231 breast cancer cell lines were cultured in serum-free, stem cell factor-supplemented medium under non-adherent conditions and passaged to generate 3rd generation spheroids. The spheroids were co-cultured with fetal lung fibroblast (FLF) cells before orthotopic heterotransplantation into the mammary fat pads of athymic nude rats. Excised xenografts were assessed histologically by H&E staining and immunohistochemistry for breast cancer marker (ERB1), proliferation marker (Ki67), mitotic marker (pHH3), hypoxia marker (HIF-2α), CSC markers (CD47, CD44, CD24, and CD133), and vascularization markers (CD31, CD34). Breast cancer cells cultured in stem cell factor supplemented medium generated 3D spheroids exhibited increased stem-like characteristics. The 3D stem-like spheroids co-cultured with FLF as supporting stroma reproducibly and efficiently established orthotopic breast cancer xenografts in the athymic nude rat.

## 1. Introduction

Breast cancer remains the primary cause of cancer-associated death in women worldwide [1]. The heterogeneous nature of breast cancer and the high level of plasticity of breast cancer stem cells (CSCs) add an extra layer of complexity that complicates treatment [2]. A significant challenge to understanding the pathophysiology of the disease and potential therapeutic options is the lack of experimental models that can effectively recapitulate the heterogeneous and stem cell-driven nature of this disease [3,4].

The three-dimensional (3D) multicellular tumor spheroid model is an emerging platform to study tumor cell development, morphology, cellular motility, and multidrug resistance in vitro [5,6,7,8,9]. Spheroids mimic the in vivo microenvironment with respect to multicellular organization and extracellular matrix (ECM) deposition, both of which play a dominant role in epithelial-mesenchymal transition (EMT), metastasis, and enhanced resistance to chemotherapy [9,10,11]; thus, spheroid models are useful for screening novel anticancer drugs [12,13]. Spheroids also develop oxygen and nutrient spatial gradients, resulting in cells at various stages of progression (proliferating, quiescent, apoptotic, hypoxic, and necrotic) that more closely resemble what happens with tumors in vivo [14,15].

Many of the current methods of spheroid development, such as the hanging drop method, spontaneous spheroid formation, suspension, scaffold-based, and magnetic levitation, focus on mimicking the physical 3D structure of tumors. However, these methods are limited in how well critical in vivo elements, such as tumor-initiating cells or tumor microenvironment (TME) components, are incorporated. These methods are also limited in their ability to predict therapeutic response [16,17,18]. For example, the molecular development of spheroid formation by cancer cells may involve: (a) cell surface glycosylated proteins binding ECM proteins such as fibronectin to induce 3D cohesion [16], (b) the induction of critical oncogenic genes [17], and (c) the cell surface integrins’ interactions with the ECM and intracellular components within the cellular cytoskeleton in response to mechanical stimulation [17,18].

To properly recapitulate solid tumors, 3D spheroids need to emulate key defining characteristics such as an abnormal multicellular architecture, enhanced cell–cell contact, gradients of metabolites and oxygen tension, and heterogeneous vasculature [19]. Tumor cells in the poorly vascularized tumor center are often either quiescent or apoptotic and exhibit increased cell–cell contact in a hypoxic and acidic environment. In contrast, tumor cells in the periphery are well perfused, metabolically active, and experience significant tumor cell–ECM contact [20]. Perhaps, most significant is the presence of highly malignant CSCs, which are critically linked to chemoresistance, metastasis, and tumor progression in vivo [21,22].

Recently, we reported that bronchial carcinoid spheroids grown under stem cell factor-enriched medium in spheroid-promoting conditions compared to 2D monolayer cultures were enriched for the expression of the well-characterized stem cell markers ALDH1, CD44, Oct-4, Sox-2, and Nanog [23]. These 3D multicellular stem-like spheroids also had a tumor-initiating cell population with significantly increased tumorigenicity as xenografts. Thong et al. [24] identified critical aspects of stem cell biology in the formation of spheroids, including their proliferation potential, their ability to self-renew, and their ability to differentiate into downstream progeny. The single stem cell proliferation capacity can be assessed by spheroid size, which can be serially passaged to assess stem cell self-renewal capacity over time.

The present report provides evidence for developing 3D multicellular stem-like properties of breast cancer cells under stem cell factor-enriched medium and spheroid-promoting conditions. These spheroids possess enhanced proliferation potential, self-renewal, differentiation into downstream progeny, and increased tumorigenicity as xenografts in the athymic nude rat model. Due to breast cancer cell lines having differential properties to form mammospheres, we evaluated three different cell lines: MCF-7, ZR-75-1, and MDA-MB-231. It is noteworthy that in a previous report where Iglesias et al. studied the capability of eleven breast cancer cell lines to form mammospheres and proliferate in a non-adherent state [25], MCF-7 could be passaged to develop mammospheres (as could T47D, BT474, MDA-MB-436, and JIMT1) but MDA-MB-231 formed only temporary aggregates (also with SKBR3, MDA-MB-468, and MDA-MB-435 breast cancer cell lines), with a dramatic reduction in cell viability upon a second passage. In our study, MCF-7, ZR-75-1, and MDA-MB-231 formed spheroids exhibiting increased stemness characteristics when cultured in stem cell factor-supplemented media. When the spheroids were co-cultured with human fetal lung fibroblast cells (FLFs) as a stromal component, they yielded a reproducible and high take rate efficiency in establishing orthotopic xenografts exhibiting a CSC phenotype, hypoxia, and increased vascularization in the athymic nude rat model of human breast cancer.

## 2. Materials and Methods

### 2.1. Cell Lines

MCF-7, ZR-75-1, and MDA-MB-231 human breast cancer cell lines were obtained from American Tissue Culture Collection ATCC, 10801 University Boulevard, Manassas (VA) 20110-2209 USA). MDA-MB-231 is a triple negative breast cancer cell line (ER−/PR−/HER2/neu−) and the most difficult to treat, while MCF-7 and ZR-75-1 (ER+/PR+/HER2/neu−) are amenable to hormone therapy [26]. Human FLF cells were obtained from ATCC.

### 2.2. 2D Monolayer Cell Culture

For routine 2D monolayer cell culture, cells were cultured in RPMI-1640 medium (R8758; Sigma Aldrich Canada, 2149 Winston Park Dr, Oakville, ON L6H 6J8, Canada) supplemented with 10% fetal bovine serum (FBS) and 0.5% penicillin/streptomycin at 37 °C with 5% CO_2_. Fresh medium was added every other day. Cells were harvested by washing 80% confluent flasks with phosphate-buffered saline (PBS; 10-010-049; Gibco) and adding 0.05% trypsin EDTA (25300054; Gibco) to detach cells. Cells were harvested and counted, and 4 × 10^7^ cells/mL were resuspended in culture medium.

### 2.3. Generation of 3D Multicellular Stem-Like Human Breast Tumor Spheroids

Human breast cancer cell lines were cultured in non-adherent conditions at a density of 5 × 10^6^ cells/mL in 75 cm^2^ 3% agarose-coated cell culture flasks in serum-free RPMI-1640 medium (R8758; Sigma Aldrich Canada) supplemented with 30 ng//mL of basic fibroblast growth factor (bFGF; 100-18B; PeproTech Canada, Office Servicing Canada, 5 Cedarbrook Drive, Cranbury, NJ, USA), 30 ng/mL of epidermal growth factor (EGF; AF-100-15; PeproTech Canada) 4 ng/mL of insulin (91077C; Sigma-Aldrich Canada), 0.5% bovine serum albumin (BSA; 15561020; Thermo Fisher Scientific), and 0.03% B27 (17504044; Thermo Fisher Scientific, Fisher Scientific Company, Ottawa, ON, Canada), as previously reported [23]. Once spheroids started to form, the growth factor-supplemented culture medium was replaced every other day. For passaging, 0.05% trypsin was added to the cells and gently centrifuged (1200 rpm for 4 min), and the pellet was gently dissociated with a blunt-end 16-gauge needle into single-cell suspensions. This process was repeated to obtain a 3rd generation passage of 3D multicellular spheroids.

### 2.4. Methylcellulose Colony Formation Assay

The methylcellulose colony formation assay was used to assess clonogenic potential as previously described [27]. MCF-7, ZR-75-1, and MDA-MB-231 2D culture cells and 3D multicellular spheroids were trypsinized, and single cells were resuspended in a medium consisting of 40% methylcellulose (Methocult; STEMCELL Technologies Canada Inc., Vancouver, BC, Canada) containing 49% RPMI-1640, 10% FBS, and 1% penicillin/streptomycin. Single-cell suspensions were gently vortexed and distributed into non-adherent 35-mm tissue culture dishes with a blunt-end 16-gauge needle at a concentration of 3 × 10^3^ cells/dish. Each condition was plated in triplicate, and samples were incubated at 5% CO_2_ and 37 °C. After two weeks, the number of formed colonies were counted using a grading dish and photographed on a phase-contrast microscope (10×). Clonogenic potential was measured as the average number of colonies per 3 dishes per number of cells seeded for each cell line for the 2D monolayer and 3D spheroids.

### 2.5. Riboflavin Autofluorescence and HIF-2α Immunofluorescence Labeling

2D monolayer breast cancer cells and 3rd generation spheroids were grown on 12-mm glass coverslips in a 24-well plate until 75% confluent and incubated with riboflavin (1:50) in 1xPBS for 1 h, washed 3 times with PBS, and fixed with 80% methanol at −20 °C and mounted in PBS/glycerol (1:1). Spheroids were collected by centrifugation, washed in cold PBS, and gently flattened under a coverslip to expose the interior and fixed with 80% methanol at −20 °C and mounted in PBS/glycerol (1:1). Cells were counterstained by DAPI to visualize the nuclei. Coverslips were examined by epifluorescence microscopy (λex/λem = 450/525 nm).

For HIF-2α immunofluorescence labeling, 2D monolayer cells were cultured on glass coverslips and flattened spheroids were permeabilized with 0.3% Triton X-100 for 3 min. The cells were washed with PBS and blocked with 5% BSA in 0.1% PBS-Tween (PBST) for 30 min at room temperature. The rabbit anti-HIF-2α (1:100) primary antibody was made up in 5% BSA-PBST, transferred onto the coverslips, and incubated at 4 °C overnight. The cells were washed with PBS and incubated with AlexaFluor 488 conjugated secondary antibody (1:100) in 5% BSA in 0.1% PBST for 1 h at room temperature. Samples were also incubated with the secondary antibody alone as the isotype control. The cells were washed with PBS and incubated with DAPI nuclear counterstain (1:10,000) for 10 min at room temperature. The cells were visualized, and fluorescence images were obtained using a fluorescence microscope (10× and 40× magnification) (Nikon DXM1200 digital camera, 331 Norton Eclipse software version 6.1).

### 2.6. Flow Cytometry

Adherent cells and dissociated spheroid cells were obtained after trypsinization, washed, and fixed with 4% paraformaldehyde (PFA; Marivac-Canemco Inc., 1 chemin du lac Chevreuil, Gore, QC, Canada) in PBS, and 3 × 10^6^ cells/mL were permeabilized with 0.1% Triton-X in PBS, washed twice with PBS, and blocked with a 5% BSA/PBS solution for 1 h at room temperature. Cells were incubated with primary antibodies CD24 (0.25 μg per 10^6^ cells/100 μL; ab64064), CD44 (1:300; ab51037), CD47 (1:100; ab108415), and CD133 (1:300; ab19898) in 5% BSA/PBS at 4 °C overnight. Cells were subsequently washed three times with PBS and incubated with secondary mouse (1 μg per 10^6^ cells/100 μL; ab99772) or rabbit (1:50; ab98488) antibody in 5% BSA/PBS, for 1 h at room temperature. Cells were washed and analyzed on a BD FACSCAN flow cytometer.

### 2.7. AlamarBlue Cytotoxicity Assay

Drug-induced cytotoxicity was determined using the AlamarBlue assay, as previously described [23]. 3 × 10^3^ monolayer breast cancer cells and 3rd generation spheroids were plated in 96-well plates and treated with cisplatin (0–40 μM) or DMSO control (0.2 × 10^−5^ μM) for 72 h. The AlamarBlue agent (BUF012B; Bio-Rad Laboratories (Canada) Ltd., Mississauga, ON, Canada) was added as 10% of the total volume to each well for 4 h. Fluorescence was measured using the SPECTRAmax Gemini Spectrophotometer (λex/λem = 540/590 nm).

### 2.8. Trypan Blue Exclusion Viability Assay

The trypan blue exclusion assay was used to assess cell viability. Following trypsinization, cells were incubated with trypan blue (Wisent Inc., Saint-Jean-Baptiste, QC, Canada) for 10 min. The number of trypan blue positive cells per total cells per microscopic field (total 4 fields per condition) were counted and calculated to obtain percent cell viability.

### 2.9. Xenograft Animal Study

Fifteen healthy 6-week-old female athymic nude rats (Harlan Laboratories Inc., 8520 Allison Pointe Blvd #400, Indianapolis, USA) were used in this study, and all efforts were made to minimize distress. All procedures were approved by the Hospital for Sick Children Animal Care Committee (protocol #22918, dated 1 May 2019) and conducted in accordance with the Animals for Research Act of Ontario and the Guidelines of the Canadian Council on Animal Care.

Breast cancer monolayer cells and 3rd generation spheroids were co-cultured with human FLFs prior to injection into the nude rat to include a stromal cell component. FLF cells were cultured at a density of 2 × 10^6^ cells/mL in adherent and non-adherent conditions (as described above) and maintained in culture for 24–72 h. Cells were examined every 24 h under light microscopy using the trypan blue assay to assess viability. An optimal time of 24 h to maintain maximum viability of FLF cells under adherent and non-adherent culture conditions was previously determined. With longer culture times, FLF had altered cell morphology and reduced viability. To determine the optimal ratio of breast cancer cells and FLF, 1–5 × 10^6^ cells/mL of monolayers and 3rd generation spheroids of each tumor cell line were co-cultured with 1:1, 1:2, 1:3, and 1:5 FLF cells. Cultures were examined every 24 h by light microscopy, and the trypan blue assay was used to assess viability. A 1:3 of FLF (1 × 10^6^ cells/mL) and breast cancer cells in adherent and non-adherent conditions (3 × 10^6^ cells/mL) was previously determined as the optimal condition for all cell lines.

A 1:3 ratio of FLF and 3rd generation spheroids co-cultured for 24 h was orthotopically transplanted at a total of 5 × 10^6^ viable cells in Matrigel (1:1; 356234, BD Biosciences, San Jose, CA, USA) using a 28-gauge needle into the mammary fat pad of nude rats (*n* = 6). In the comparison group (*n =* 9), a 1:3 ratio of FLF and monolayer cells co-cultured for 24 h was inoculated as a total of 5 × 10^6^ viable cells in Matrigel and orthotopically transplanted into the mammary fat pad of nude rats. All animals were anesthetized with 1.5% isoflurane before injections. Tumors were excised before they reached the permitted endpoint (1.5 cm in diameter) or when animals showed signs of distress (e.g., severe dehydration), whichever came first. Samples were fixed in 10% neutral buffered formalin at room temperature and processed for histopathology.

### 2.10. Magnetic Resonance Imaging (MRI)

Rats were imaged weekly on a 3-Tesla Clinical MRI scanner (Achieva 3.0T TX, Philips Medical Systems, Best, Netherlands), beginning at 7 days after tumor cell inoculation, using an eight-channel transmit/receive wrist coil for signal detection. Rats were first induced with 2% isoflurane in pure oxygen (2 L/min flow rate) and then maintained on 1.5% isoflurane during imaging. Rats were placed prone within the coil, resting on top of a water blanket maintained at 36 °C (HTP-1500; Adroit Medical Systems, Loudon, TN, USA). High-resolution anatomical T2 weighted turbo spin-echo scans were acquired. The T2 weighted turbo spin-echo scan used a 2D acquisition with the following parameters: repetition time = 4000 ms, echo time = 75 ms, echo train length = 16, number of spinal averages = 2, 100 mm field-of-view, 20 1-mm thick slices, and 0.6 × 0.6 in-plane resolution. Tumors were detected as a hyperintense (bright) signal on T2-weighted images.

### 2.11. Histology and Immunohistochemistry

Fixed tumors were embedded in paraffin, sectioned at 5 μm, and sections were transferred onto glass slides, deparaffinized through xylene and graded alcohols into water, and stained with hematoxylin and eosin (H&E). For routine immunohistochemical (immunoperoxidase) labeling, antigen retrieval was performed in 10 mM sodium citrate buffer (pH 6) by heating in a microwave oven for 10 min. The sections were cooled for 20 min at room temperature and then incubated in 3% hydrogen peroxide in water for 10 min to block endogenous peroxidase activity. The sections were washed with 10× deionized water for 5 min and incubated with 10% goat or horse serum in PBST for 30 min to block non-specific binding. The sections were then incubated with the appropriate primary antibodies for breast cancer marker (ERβ1) (1:200; NBP1-41201; Novus Biologicals, Toronto, ON, Canada), stem cell markers CD24 (1:50, ab64064) CD44 (1:50; ab51037), CD47 (1:50; ab175388) and CD133 (1:200; ab19898), vascular markers CD31 (1:50; ab28364) and CD34 (1:2500; ab81289), HIF-2α (1:200; ab113642), proliferation marker (Ki67) (1:250; ab15580), and mitotic marker phospho-Histone H3 (PHH3) (1;100; MA5-15220; Thermo Fisher Scientific) in 5% BSA/PBST, and incubated at 4 °C overnight. The sections were washed with PBS 10 times for 5 min each, incubated with the appropriate secondary antibodies, and incubated for two minutes with DAB (3,3′- diaminobenzidine; Vector Laboratories, Inc., Burlingame, CA, USA). The sections were washed with deionized water and counterstained with hematoxylin. Once counterstaining was complete, the slides were dehydrated and mounted. All stained sections were assessed on a light microscope (Olympus BX60).

### 2.12. Statistical Analysis

The data are presented as mean ± standard error of the mean (SEM). Statistical analysis of variance was run on triplicate experiments using GraphPad Prism 5.0 software (GraphPad Software, San Diego, CA, USA) using an unpaired 2-tailed Student’s t-test at 95% confidence.

## 3. Results

### 3.1. Breast Cancer Cell Lines Form 3D Multicellular Spheroids on Agarose in Stem Cell Growth Factor Enriched Conditioned Medium after 3rd Generation Passage

Human breast cancer cell lines, MCF-7 and ZR-75-1 (ER+/PR+/HER2/neu−), and the triple-negative MDA- MB-231 cell line (ER−/PR−/HER2/neu−) were grown as spheres on agarose-coated flasks in an enriched medium for stem-like cells, as described for the small cell lung cancer (SCLC) cell line H446 [28] and bronchial carcinoid cell lines (H727 and H720) [23]. The stem cell factor culture medium containing basic fibroblast growth factor, epidermal growth factor, insulin, bovine serum albumin, and B27 growth factors was replaced every other day. To obtain 3rd generation passage of 3D multicellular spheroids, the cultures were trypsinized into single-cell suspensions and cultured again in stem cell factor-supplemented medium, as depicted in Figure 1A. MCF-7, ZR-75-1, and MDA-MB-231 cells were able to form spheroids (SP) in stem-cell-factor-supplemented culture conditions when plated from monolayer culture (Figure 1B). All three cell lines robustly produced spheroids under the specified culture conditions. In particular, MCF-7 and MDA-MB-231 formed spheroids larger in size and number compared to the ZR-75-1 cell line (Figure 1C).

### 3.2. Clonogenic Potential of Breast Cancer Cell Lines Forming 3D Multicellular Spheroids

We previously reported that 3D multicellular stem-like bronchial carcinoid spheroids [23] and another report for human SN12C renal cell carcinoma cell line [29] generated in stem-cell culture medium had increased clonogenic potential. Here, the colony formation assay was used to measure the clonogenic potential of 3D multicellular breast cancer spheroids, compared to their monolayer counterparts over 14 days (Figure 2). The 2D monolayer cells and the 3D breast cancer spheroids were dissociated and seeded as single cells in methylcellulose-containing medium. The 2D monolayers formed a small number of small- to medium-sized colonies, with a few larger ones with all three breast cancer cell lines. In contrast, the dissociated cells from the 3rd generation spheroids formed a greater number of medium- and larger-sized spheroids. Collectively, the 3rd generation spheroids formed a significantly greater number of spheroids compared to their monolayer counterparts. As shown in Figure 2A, the MDA-MB-231- and MCF-7-derived spheroids formed large colonies, while the ZR-75-1-derived spheroids formed smaller ones; furthermore, ZR-75-1 colonies were fewest in number. The data also demonstrate that the clonogenic potential of 2D monolayer versus spheroids increased from 13 to 25 ± 0.15 (MCF-7), 4 to 12 ± 0.04 (ZR-75-1), and 18 to 32.33 ± 0.33 (MDA-MB-231) (Figure 2B). Since greater colony size infers a greater proliferative capacity of single cells, this assay suggests the presence of cancer stem cells, with the triple-negative breast MDA-MB-231 cells possessing the greatest tumor-forming potential in vivo.

### 3.3. Riboflavin Uptake in 3rd Generation 3D Breast Cancer Spheroids Confirms Stem Cell Characteristics

Reports have shown that stem cells exhibit remarkable riboflavin uptake via ATP-dependent ABCG2 transporters in cytoplasmic vesicles [30,31]. Riboflavin, also known as vitamin B2, is an easily absorbed micronutrient with a critical role in maintaining a wide variety of cellular characteristics, including energy metabolism and the metabolism of fats, ketone bodies, carbohydrates, and proteins. The intrinsic autofluorescence properties of riboflavin are λex/λem = 270/525 nm, 370/525 nm, and 450/525 nm. To confirm the stem-like cell potential of the 3rd generation of breast cancer spheroids, the autofluorescence of riboflavin was measured in the 2D monolayer cells and 3rd generation 3D spheroids (Figure 3). MCF- 7, ZR-75-1, and MDA-MB-231 monolayers and 3rd generation spheroids were incubated with excess riboflavin and assessed for riboflavin autofluorescence (Figure 3A,C,E). Riboflavin uptake measured via autofluorescence is strikingly associated with spheroids. While breast cancer cells cultured as monolayers did not exhibit appreciable fluorescence from riboflavin accumulation, the 3rd generation spheroids displayed differential fluorescence. In line with the clonogenic assay, MDA-MB-231 SP displayed the most significant riboflavin uptake, 90 ± 0.66% greater autofluorescence compared to MDA-MB-231 PA, followed by MCF-7 SP and ZR-75-1 SP, which demonstrated a 55 ± 0.66% and 43 ± 0.11% greater riboflavin uptake, respectively, compared to their corresponding monolayer (*n* = 3, *p* < 0.001) (Figure 3B,D,F). The increased riboflavin accumulation in the 3rd generation spheroids suggests that anchorage-independent tumor cells cultured in stem cell factor-supplemented media select a unique stemness phenotype, which is suggestive of a greater tumorigenic potential. Both the colony-forming potential and riboflavin uptake assessment support the presence of a greater stem-like tumor-initiating cell population in the 3rd generation spheroids grown in enriched stem-cell culture medium containing basic fibroblast growth factor, epidermal growth factor, insulin, bovine serum albumin, and B27 growth factors.

### 3.4. HIF-2α Expression in 3D Multicellular Stem-like Spheroids of MCF-7, ZR-75-1, and MDA-MB-231 Breast Cancer Cells

Previously, we reported on the role of HIF-2α in suppressing p53, resulting in a state of “enhanced stemness” in human embryonic stem cells [32]. In breast cancer, HIF-2α is overexpressed in 29% of primary lesions and 69% of breast metastases but is not expressed in benign tumors [33], because hypoxia- sensitive gene expression is induced in rapidly growing tumors with a hypoxic core [34,35]. 3D tumor spheroid cultures, which can recapitulate the physiology of the in-vivo tumor microenvironment, also display gradients in oxygen as well as in pH and nutrients [14,36]. Here, we investigated whether our 3rd generation spheroids have greater HIF-2α expression than their monolayer counterparts. MCF-7, ZR-75- 1, and MDA-MB-231 monolayers and 3rd generation spheroids were immunostained for HIF-2α expression (Figure 4). Consistent across all three cell lines, 3rd generation spheroids had a significantly greater expression of HIF-2α compared to their monolayer counterparts. Furthermore, MDA-MB-231 SP had the highest level of HIF-2α expression (84 ± 0.33), followed by MCF-7 SP (64 ± 0.35), and ZR-75-1 SP (14 ± 0.66) (*n* = 6, *p* < 0.001). These results are in line with the previous riboflavin and clonogenic colony formation results, suggesting that 3rd generation spheroids have the highest proportion of cancer stem cells, with MDA-MB-231 being the most aggressive.

### 3.5. 3D Multicellular Stem-Like Spheroids of MCF-7, ZR-75-1, and MDA-MB-231 Breast Cancer Cells Express CD47, CD44, CD24, and CD133 Markers of Cancer Stem Cells

To determine whether 3rd generation spheroids select for cells with CSC properties, fluorescence-activated cell sorting (FACS) analyses were used to assess the expression of CD47, CD44, CD24, and CD133 CSC markers in MCF-7, ZR-75-1, and MDA-MB-231 2D monolayers and 3rd generation spheroids (Figure 5). There was a significantly greater expression of the CSC markers in the 3rd generation spheroids versus the 2D monolayer counterparts across all cell lines, at an order of 5-8-fold for MCF-7 and MDA-MB-231, and approximately 2-fold for ZR-75-1. MCF-7 SP expressed 23 ± 0.35% more CD47, 27 ± 0.53% more CD44, 24 ± 0.56% more CD24, and 15 ± 0.70% more CD133, compared to MCF-7 PA (Figure 5A,B). ZR-75-1 SP expressed 1 ± 0.96% more CD47, 3 ± 0.41% more CD44, 3 ± 0.01% more CD24, and 2 ± 0.45% more CD133 (Figure 5C,D). MDA-MB-231 SP expressed 18 ± 0.56% more CD47, 28 ± 0.70% more CD44, 26 ± 0.25% more CD24, and 16 ± 0.02% more CD133, compared to MDA-MB-231 PA (Figure 5E,F). ZR-75-1 was found to express the lowest level of CSC markers. CD44 and CD24 were the highest expressed across all three cell lines (Figure 5G), in line with previous reports [37,38,39]. The FACS analysis data corroborate the riboflavin data, suggesting that riboflavin uptake may identify earlier CSC progenitors [31]. Collectively, the data demonstrate that stem cell factor-supplemented 3D spheroid culture promotes the acquisition of a stemness phenotype.

### 3.6. 3D Multicellular Stem-Like Spheroids of MCF-7, ZR-75-1, and MDA-MB-231 Breast Cancer Cells Display Increased Chemoresistance to Cisplatin

The acquisition of stem-like characteristics of cancer cells is typically an advent of the survival response following chemotherapeutic treatments [40,41,42,43]. In particular, HIF-2α expression has been reported to be associated with resistance to chemotherapy [44]. The data in Figure 4 demonstrates that 3rd generation breast cancer spheroids have increased expression of HIF-2α. To assess whether the 3rd generation spheroids show greater chemoresistance than their monolayer counterparts, MCF-7, ZR-75-1, and MDA- MB-231 monolayer cells and 3rd generation spheroids were exposed to chemotherapeutic agent cisplatin (0–40 μM) for 72 h. Cell viability was measured using the AlamarBlue cytotoxicity assay. The data depicted in Figure 6 demonstrate that cisplatin cytotoxicity is concentration-dependent across all cell lines, with the spheroids displaying greater chemoresistance (LC50 values) than their monolayer counterparts. More specifically, MCF-7 SP had an LC50 of 32 compared to MCF-7 PA, which had an LC50 of 23, ZR-75-1 SP LC50 22 vs. 17 for ZR-75-1 PA, and MDA-MB-231 SP LC50 27 vs. 20 for MDA-MB-231 PA. These results suggest that 3rd generation 3D multicellular stem-like spheroids are more chemoresistant than their 2D counterparts, confirming their acquisition of stem-like characteristics.

### 3.7. 3D Multicellular Stem-Like Breast Cancer Spheroids Co-Cultured with Human Fetal Lung Fibroblasts Successfully Establish Orthotopic Human Breast Tumors in Athymic Nude Rats with Increased Tumorigenicity

We observed that the 3rd generation 3D multicellular breast cancer spheroids have a significant cancer stem-like cell population. Based on these observations, we posited an increased tumorigenic potential in vivo. To test this hypothesis, the in vivo tumorigenic activity of the 3rd generation 3D multicellular breast cancer spheroids was investigated in an athymic nude rat model of human breast cancer. As shown in Figure 7, cells derived from the 3rd generation spheroids were admixed with human fetal lung fibroblasts (FLF) to establish an extracellular stromal component and orthotopically injected into the mammary fat pad in Matrigel (*n* = 6 nude rats). The addition of human fetal lung fibroblasts provided growth factors, chemokines, and ECM components, maintaining structural integrity and facilitating tumor angiogenesis [45,46]. To determine if these FLF supplemented spheroids would increase the orthotopic tumor take rate, control cohort rats were inoculated with 2D monolayer cancer cells co-cultured with FLF (*n* = 9).

Tumor growth was accelerated in all animals inoculated with spheroid cells, with tumors appearing on MRI by day 7 (Figure 8A). Tumors were recognized by their hyperintense, bright contrast on T2 (transverse relaxation time), typically due to the high water content found in most tumors. Most animals were sacrificed between day 14 and day 21 due to severe dehydration, likely due to tumor burden. Only one ZR-75-1 SP inoculated rat survived until the tumor reached the designated endpoint. In line with the in vitro data, MCF-7 and MDA-MB-231 spheroids formed the largest tumors over the 10-day growth period (Figure 8B). In contrast, the rats inoculated with the 2D monolayer cells did not develop tumors and were monitored weekly by MRI up to day 105 after inoculation (data not shown). There was no evidence of tumor development on imaging or gross dissection. The take-rate of animals inoculated with spheroid cells was 100%, while animals inoculated with 2D monolayer cells had no take-rate.

H&E staining of the xenografts revealed cancer cells with some pyknotic or fragmented nuclei amongst mainly viable epithelioid cell islands with a glandular-like arrangement but all distinctly different in their histological appearance (Figure 8C). To confirm the breast cancer phenotype, xenografts were immunostained for the ERβ1 marker (Figure 8D). ERβ1 has been previously reported to be a prognostic breast marker [47,48]. There were 163 ± 0.33 ERβ1-positive immunostained cells/field for MCF-7 SP, 70 ± 0.28 cells for ZR-75-1, and 175 ± 0.11 for MDA-MB-231. In line with previous results, ZR-75-1 spheroid xenografts displayed less ERβ1 labeling than MCF-7 and MDA-MB-231 spheroids, suggesting a small population of stem-like cancer cells (Figure 8D).

### 3.8. Characterization of Breast Cancer Cell Line-Derived Spheroid Xenografts in Athymic Nude Rats

Xenografts were immunostained for Ki67 and pHH3 to assess proliferative and mitotic properties, respectively (Figure 9). Ki67 has been previously reported as a prognostic marker for breast cancer and is strongly associated with other prognostic markers [49,50]. As expected, MCF-7 SP and MDA-MB-231 SP xenografts displayed a greater number of Ki67 immunopositive cells (350 ± 0.11 cells and 224 ± 0.14 cells, respectively) compared to ZR-75-1 SP (117 ± 0.25 cells) (Figure 9A,B). Mitotic marker phospho-histone H3 (pHH3) can enhance the prognostic value of Ki67 and act as an improved proliferation marker in breast cancer [51]. Immunostaining for pHH3 followed the same pattern as Ki67, with the greatest expression in MCF-7 SP (83 ± 0.65 cells/field) and MDA-MB-231 SP (59 ± 0.15 cells/field) xenografts and less expression in ZR-75-1 SP (28 ± 0.30 cells/field) xenografts (Figure 9 C,D). Collectively, the immunostaining data confirm that the breast cancer xenografts established in nude rats from 3rd generation spheroids displayed elevated proliferative and mitotic fractions associated with a poor prognosis. Proportionally, the two markers had similar expression profiles across the three different xenografts, with ZR-75-1 SP xenografts having a lower expression, suggesting that it is a slower growing tumor compared to MCF-7 SP and MDA-MB-231 SP tumors.

To assess whether xenografts established from cells from 3rd generation spheroids exhibit the same characteristics as spheroids cultured in stem cell factor-supplemented culture media in vitro, we examined immunoreactivity for hypoxic marker HIF-2α (Figure 10) and stem cell markers CD47, CD44, CD24, and CD133 (Figure 11A). As previously shown, 3D spheroids expressed elevated HIF-2α, which is expected to be in line with tumors in vivo. Tumor xenografts immunostained for HIF-2α followed the same trends as 3D spheroids in vitro, with MCF-7 SP and MDA-MB-231 SP displaying the greatest number of HIF-2α immunopositive cells/field (163 ± 0.25 and 175 ± 0.15, respectively) and ZR-75-1 SP having the lowest number of HIF-2α immunopositive cells/field (104 ± 0.1) (Figure 10). Concerning the evaluation of the stemness markers CD47, CD44, CD24 and CD133, the xenografts did not match the expression profile observed in the in vitro spheroids (Figure 5). Although all three spheroid xenografts displayed a degree of stemness expressions, the MCF-7 SP and MDA-MB-231 SP have greater expression and ZR-75-1 SP have a lower expression (Figure 11B). All three xenografts expressed high levels of CD133. MCF-7 SP and MDA-MB-231 SP xenografts expressed mainly CD24 and CD133, while ZR-75-1 SP xenografts had CD44 and CD133. This variation in stem cell marker expression across the three different xenografts likely reflects the differential stem-like properties.

Due to the interplay between hypoxia, stemness, and angiogenesis, we examined the degree of vascularization of the 3D stem-like spheroid-derived breast tumor xenografts by immunostaining for CD31 and CD34, reported to be markers of angiogenesis and indicative of a poor prognosis [21,52]. All three xenografts displayed a moderate level of vascularity as observed by CD31 and CD34 immunostaining (Figure 12). MCF-7 SP and MDA-MB-231 SP xenografts had greater CD31 immunopositive cells per field (143 ± 0.65 and 107 ± 0.13, respectively) than CD34 immunopositive cells per field (63 ± 0.28 and 78 ± 0.34, respectively). For ZR-75-1 SP, CD31 and CD34 expression were equivalent, with 49 ± 0.30 immunopositive cells per field and 47 ± 0.25 immunopositive cells per field, respectively (Figure 12). These data suggest that MCF-7 SP and MDA-MB-231 SP have a higher number of endothelial progenitors, correlating with a faster-growing tumor, in line with the previous data.

## 4. Discussion

Tumor spheroids show promise as a model for investigating the growth and development of tumors, cancer cell motility, and the efficacy of chemotherapeutic agents. The present report provides evidence for the critical role of stem cell properties expressed in tumors to form multicellular tumor spheroids in vitro and in vivo xenograft tumors. Importantly, establishing breast tumor xenografts in the athymic nude rat model is particularly challenging. In this study, we developed an efficient, reproducible method of generating 3D stem-like human breast tumor xenografts in athymic nude rats.

Xenografting typically involves implanting tumor fragments or tumor cells that have been cultured in a traditional 2D monolayer environment. With this approach, a recent study showed that the take rate varied from 0% when human MDA-MB-231 breast cancer cells were injected subcutaneously in nude rats to 25% when Matrigel was co-injected [52]. An alternative to 2D monolayer culturing is the 3D spheroid culture method. Multicellular spheroids are important in vitro models as they can mimic the functional and architectural characteristics of in vivo tissues [7]. For example, they enable complex cell-cell interactions and establish gradients (such as nutrients, oxygen, and metabolites) as a result of barriers to diffusion imposed by the spheroid. In cancer biology, spheroids are an important model for drug discovery and are commonly used in the in vitro setting to understand cell-stromal interactions and mechanisms of cancer metastasis [53]. To use spheroids directly in vivo is a new approach. Such reports are rare, but a recent study reported an interesting finding that mouse tumor xenografts from spheroid injections were larger than those from adherent cells [54]. The idea that 3D spheroids may improve establishment of tumor xenografts in rats, however, has not been fully explored.

As shown in Figure 13, we accomplished this by first culturing the breast cancer cells as 3D spheroids in enriched stem cell factor-supplemented medium under non-adherent conditions from monolayers and passaged repeatedly to reach 3rd generation spheroids to enhance the stem cell population. The findings revealed more significant cancer stem cell properties and tumor-initiating cells. Before injection into the mammary fat pad of the rat, we co-cultured 3rd generation 3D stem-like breast spheroid cells with human fetal fibroblasts as components of the supporting stroma, resulting in the spheroid xenografts recapitulating tumor-stromal organization and architecture. Using our novel stem cell factor-supplemented 3D spheroid culture, we were able to successfully establish tumorigenic orthotopic xenografts using three different human breast cancer cell lines: MCF-7, ZR-75-1, and MDA-MB-231, with an exceptional success take rate of 100%. We have previously reported on the success of this approach in generating tumorigenic orthotopic xenografts of bronchial carcinoid in NOD/SCID mice, where spheroids admixed with FLF had a 40% greater take rate compared to just spheroids alone (unpublished data) [23].

The 3rd generation 3D breast tumor spheroids displayed a stemness phenotype, as assessed by the methylcellulose clonogenic assay, riboflavin uptake, HIF-2α and CD47, CD44, CD24, and CD133 CSC marker expression. The stemness phenotype was closely recapitulated in the corresponding xenografts, suggesting that tumor growth and tumorigenicity are driven by the presence of CSCs or tumor-initiating cells that are selected for under stem cell factor-supplemented culture conditions [55].

It is noteworthy that riboflavin accumulation was superior in the 3D spheroids compared to 2D monolayers, suggesting that the heterotransplantation of spheroids delivers a substantial number of tumor- initiating cells at the orthotopic site. Riboflavin plays an integral role in various metabolic functions and pathways [56]. Interestingly, the association of riboflavin as a marker of increased mitochondrial potential and elevated CSC activity was shown when diphenyleneiodonium chloride treatment was used to confer a mitochondrial-deficient phenotype that depleted CSCs, namely, through reduction of CD44 and CD24 [30]. Furthermore, mitochondrial and ribosomal biogenesis and high mitochondrial mass have also been reported as hallmarks of stemness and biomass accumulation in breast cancer cells that are chemoresistant and present as a means of targeting CSCs [57,58,59]. Interestingly, spheroids have been reported to produce metabolites typical for their tissue of origin, suggesting that 3D cultures bear many metabolic similarities to the original tissue, which conventionally 2D cultured cells lack [60]. However, how mitochondrial mass and riboflavin accumulation are associated with CSC markers remains to be elucidated and presents an exciting field of research.

The breast cancer stem cell markers CD47, CD44, CD24, and CD133 have all been extensively studied and reviewed [37,38,39]. Previously, reports have shown that breast cancer cell lines contain only a small subpopulation of CD44+/CD24− tumor-initiating cells [61]. Not surprisingly, FACS analysis showed that 3rd generation 3D stem-like breast tumor spheroids had a significantly greater expression of the stemness factors compared to their 2D monolayer counterparts. Among the cell lines, MDA-MD-231 SP was the most tumorigenic, with the greatest number of CSC marker positive cells, followed by MCF-7 and ZR-75-1. These data are in line with previous studies that reported that the triple-negative breast cancer cell line MDA-MB-231 spheroids displayed greater CSC genes and exhibited a more robust response to chemotherapeutics than MCF-7 spheroids [62]. Within each cell line, FACS analysis and immunohistochemistry analysis of the corresponding xenografts revealed an almost equivalent expression of CD44 and CD24. The CD44+/CD24− subpopulation has been associated with potent tumorigenic potential, acting as drivers of metastasis and therapy resistance [63,64,65]. The tumor CSC phenotype expressing CD47 has a higher degree of invasiveness and metastasis [66,67]. CD47 overexpression has been reported to enhance the CD44+/CD24− population in breast cancer, but not sufficiently to increase its stemness [68]. However, we did not find low CD24 in our spheroids or the corresponding xenografts. These data suggest that CSC marker expression levels may vary in established tumorigenic cell lines and warrants further investigation in their relationship with other CSC markers. However, due to the elevated expression of CD47 in the 3D stem-like spheroids and xenografts, we hypothesize that CD47 is a more robust CSC marker and better marker of prognosis to be a clinically validated target for cancer immunotherapy [69,70]. The co-expression of CD44 and CD47 in tumor cells may contribute to breast cancer metastasis [71,72,73]. CD133 has been directly associated with metastasis and drug resistance and presents as an additional immunotherapy target [74,75]. The elevated CD47, CD44, CD24, and CD133 in 3D stem-like spheroids and their corresponding xenografts suggest phenotypic heterogeneity in the breast CSCs with several subpopulations. Due to the crucial role of these CSC markers in migration, invasion, and metastasis, future studies should determine if the orthotopic tumors release CSCs into circulation and, if so, to characterize the phenotype of these circulating cells.

The 3rd generation spheroids and their corresponding xenografts showed robust expression of HIF-2α. HIF-2α expression has been associated with stemness, self-renewal properties, and survival genes in human breast cancer stem cells (CD24−/CD44+) exposed to hypoxia [76,77]. When studying the association of hypoxia and CSCs, Kim et al. found that the differential fates of hypoxic and non-hypoxic tumors were observed only from tumor cells isolated from the hypoxic TME in vivo and not when tumor cells were in hypoxic conditions in vitro [78]. Since hypoxia has been tightly correlated with elevated CSC marker expression, we speculate that HIF-2α sustains the CSC marker expression and stemness phenotype characteristic of the 3D spheroids. Hypoxia, namely, HIF-2α expression, also plays a crucial role in the activation of angiogenesis, providing an additional means of enhancing metastatic potential [21,79,80]. This finding reinforces the importance of the TME in orchestrating the intricate interactions that facilitate tumorigenesis. A varying degree of vascularity was observed in the xenografts, as observed by immunohistochemical analysis of CD31 and CD34 expression. Due to the xenografts attaining the maximum volume limits quickly, a minor degree of vascularity is expected, given that active proliferation was dominant.

Cancer-associated fibroblasts (CAFs) act as crucial regulators of the TME, profoundly influencing the proliferation, invasion, and metastasis of tumor cells [81,82]. Previous studies have investigated the effect of different fibroblasts on colon cancer cell proliferation, with lung fibroblasts having the greatest effect on proliferation [83]. The addition of lung fibroblasts has also been reported to result in metabolic cooperation with lung cancer cells, promoting cancer metastasis [84]. Thus, this approach to establishing xenografts in the nude rat establishes cell–cell interactions through the 3D topology of the spheroids and tumor-stromal cell interactions via the addition of fibroblasts. MRI could easily image the rapidly developed tumors, and the ease of imaging is in part due to the larger size of the rat rodent. However, the failure of any of the 2D monolayer cultured cells to establish tumors with any of the three breast cancer cell lines was unexpected, following the incorporation of Matrigel as an ECM component, as success has been reported in the past [85]. The explanation may lie in the location of the tumor cell inoculation. In theory, the mammary fat pad should provide a more favorable microenvironment for tumor initiation over other xenograft sites, such as subcutaneous inoculation, as previously reported in mouse models of breast cancer [86]. However, unless a tumor is established quickly after inoculation, the athymic nude rats may regain some immunity with time and may effectively destroy the injected foreign cells [87]. The athymic nude rats are known to be more “immune leaky” than immunocompromised mice, which results in establishing a tumor in mice over a more extended period but poses a challenge for establishing a tumor in rats relatively short period of time.

The novelty of this work lies in the efficient establishment of highly tumorigenic breast cancer orthotopic xenografts in the athymic nude rat using 3D stem-like spheroid cultures of established breast cancer cell lines. This model provides invaluable support for the athymic nude rat as a viable option for studying breast cancer progression, metastasis, and therapeutic interventions. Notably, 3D stem-like derived xenografts were successfully established using three breast cancer cell lines representing both hormone-dependent and -independent breast cancer. This xenograft approach has the added benefit of incorporating a higher tumor-initiating cell population and a critical stromal component to mimic better what is observed in vivo and improve successful heterotransplantation. Furthermore, this xenograft model offers a unique opportunity to study the heterogeneity observed in vivo using patient-derived xenografts, improving breast cancer treatments in patients with precision medicine.

## Figures and Tables

**Figure 1 cancers-13-02784-f001:**
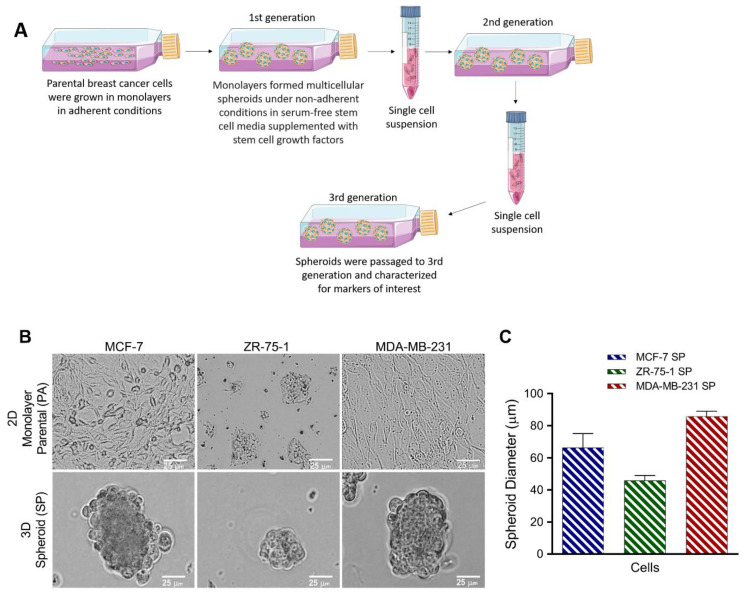
Generation of 3D multicellular breast cancer spheroids. (**A**) Human breast cancer cell lines were cultured in non-adherent conditions at a density of 5 × 10^6^ cells/mL in 75 cm^2^ agarose-coated cell culture flasks in serum-free RPMI-1640 medium supplemented with stem cell growth factors (30 ng/mL of bFGF, 30 ng/mL of EGF, 4 ng/mL of insulin, 0.5% BSA, and 0.03% B27, as previously reported [23,28]). Once spheroids started to form, the growth factor-supplemented culture medium was replaced every other day. For passaging, 0.05% trypsin was added to the cells and gently centrifuged, and the pellet was gently dissociated with a blunt-end 16-gauge needle into single-cell suspensions. This process was repeated to obtain 3rd generation passage of 3D multicellular spheroids compared with the 2D monolayer cell cultures. 2D monolayer cells and 3D 3rd generation spheroids were visualized under phase microscopy (scale bar = 25 µm). (**B**) Morphology 2D monolayer parental cells and 3rd generation 3D spheroids cultured in stem cell factor-supplemented culture medium. (**C**) Spheroid diameter of MCF-7, ZR-75-1, and MDA- MB-231 3rd generation spheroids. Abbreviations: bFGF, basic fibroblast growth factor; EGF, epidermal growth factor; BSA, bovine serum albumin; PA, parental; SP, spheroid.

**Figure 2 cancers-13-02784-f002:**
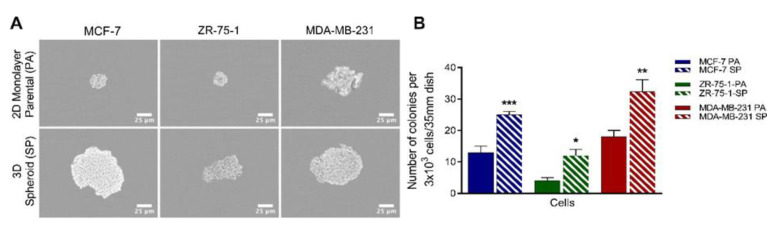
The colony-forming potential of breast cancer 2D monolayer cells and 3D spheroids using the methylcellulose colony formation assay. (**A**) MCF-7, ZR-75-1, and MDA-MB-231 breast cancer 2D monolayer cells and 3D spheroids were dissociated into single cells and seeded 3 × 10^3^ cells in 40% methylcellulose medium supplemented with 10% FBS and 1% penicillin in culture medium on 35-mm dishes for 14 days. Colonies generated were visualized under phase microscopy (scale bar = 25 µm). (**B**) Data from the methylcellulose colony assay show the average number of colonies per 3 × 10^3^ cells/35 mm dish ± SEM of three independent experiments performed in triplicates. The number of colonies formed from the SP was compared to the corresponding PA using the unpaired two-tailed t-test comparisons with 95% confidence with indicated asterisks for statistical significance. * *p* ≤ 0.05, ** *p* ≤ 0.01, *** *p* ≤ 0.001, and *n* = 3. Abbreviations: PA, 2D monolayer parental cells; SP, 3D spheroid.

**Figure 3 cancers-13-02784-f003:**
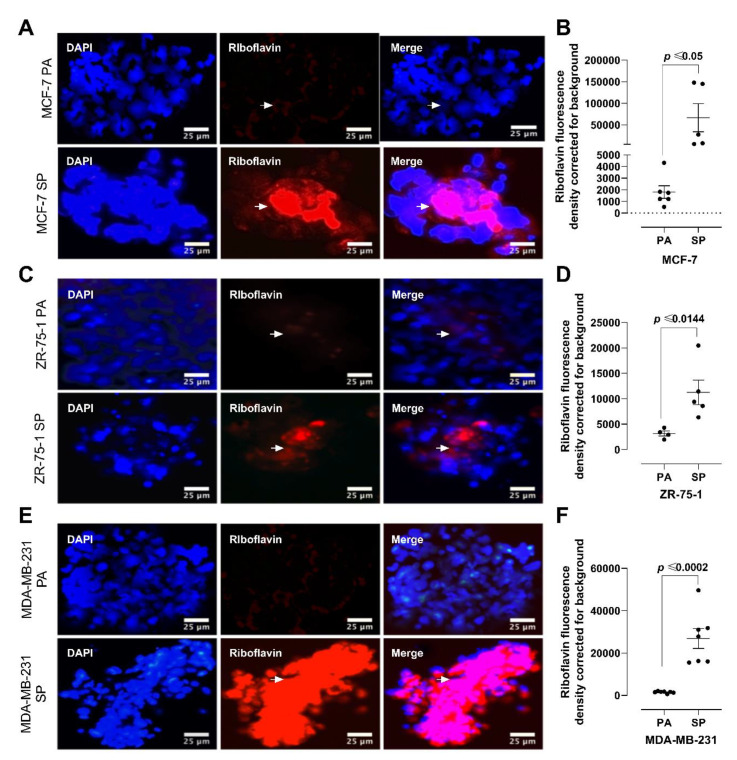
Analyses of 2D monolayer cells and 3D spheroids of MCF-7, ZR-75-1, and MDA-MB- 231 breast cancer cells for riboflavin autofluorescence using epifluorescent microscopy. (**A**,**B**) MCF-7, (**C**,**D**) ZR-75-1, and (**E**,**F**) MDA-MB-231 breast cancer cells cultured as monolayers (PA) and 3rd generation spheroids (SP) were compared for the uptake of riboflavin as detected by autofluorescence (λex/λem 540 nm/570 nm). Breast cancer PA and SP were incubated with riboflavin (1:50) for 1 h and mounted in PBS/glycerol (1:1). Cells were counterstained with DAPI to visualize the nuclei. Coverslips were examined by epifluorescence microscopy (λex/λem = 450/525 nm; scale bar = 25 μm). The results are depicted as a scatter plot, where dots represent the indicated marker values of cells obtained from the representative of 3 independent experiments of multiple cell images (*n* = 7–8). The mean fluorescence density staining corrected for background ±SEM for indicated marker values is indicated for each group. The density staining riboflavin values of each SP group were compared to the PA group using the unpaired two-tailed *t*-test comparisons with 95% confidence. Abbreviations: PA, parental; SP, spheroid; PBS, phosphate-buffered saline; DAPI, 4′,6-Diamidine-2′-phenylindole dihydrochloride.

**Figure 4 cancers-13-02784-f004:**
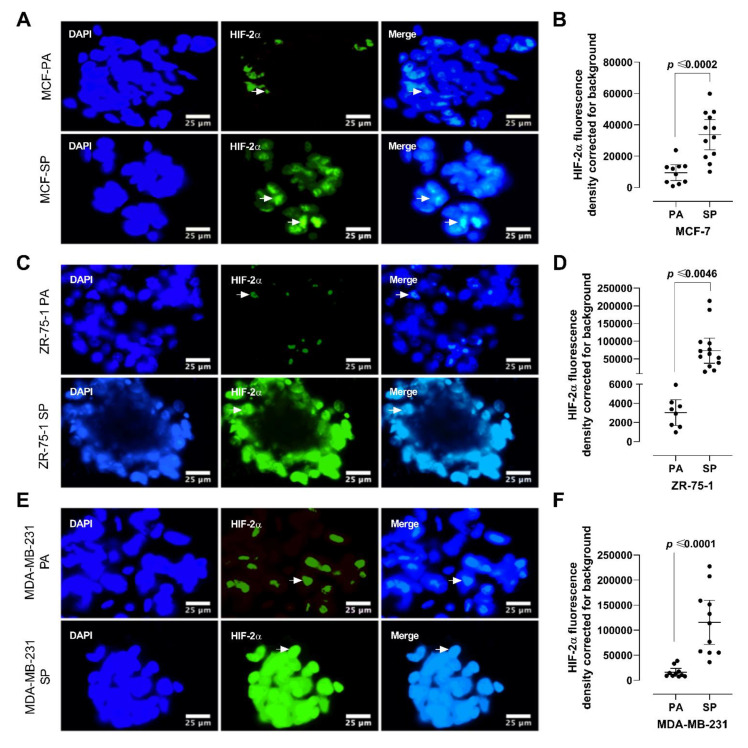
Analysis of 2D monolayer cells and 3D stem-like spheroids of MCF-7, ZR-75-1, and MDA-MB-231 breast cancer cells for HIF-2α expression using immunocytochemistry. (**A**,**B**) MCF-7, (**C**,**D**) ZR-75-1, and (**E**,**F**) MDA-MB-231 breast cancer cells were cultured as monolayers and 3rd generation spheroids and incubated with primary antibody against HIF-2α (1:100), followed with AlexaFluor 488 conjugated secondary antibody (1:100). Cells were counterstained by DAPI to visualize the nuclei. Coverslips were mounted in PBS/glycerol and examined by epifluorescence microscopy (scale bar represents 25 μm). The results are depicted as a scatter plot, where dots represent the indicated marker values of cells obtained from the representative of two independent experiments of multiple cell images (*n* = 7–16). The mean fluorescence density staining corrected for background ±SEM for indicated marker values is indicated for each group. The density staining HIF-2α values of each SP group were compared to the PA group using the unpaired two-tailed *t*-test comparisons with 95% confidence with indicated statistical significance. Abbreviations: PA, parental; SP, spheroid; PBS, phosphate-buffered saline; DAPI, 4′,6-Diamidine-2′- phenylindole dihydrochloride.

**Figure 5 cancers-13-02784-f005:**
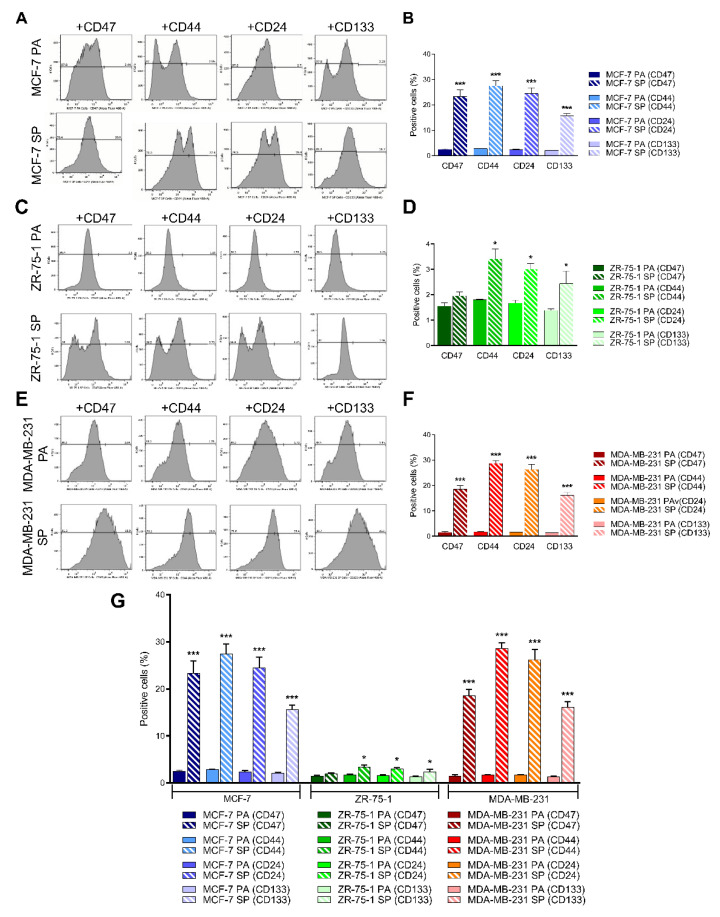
FACS analysis of stem cell markers expression on breast cancer monolayers and 3rd generation spheroids. FACS analysis of (**A**,**B**) MCF-7, (**C**,**D**) ZR-75-1, and (**E**,**F**) MDA-MB-231 monolayer (PA) and 3rd generation spheroids (SP) for CD47, CD44, CD24, and CD133. Adherent monolayer cells and spheroids were incubated overnight at 4 °C with primary antibodies against CD47 (1:100), CD44 (1:30), CD24 (0.25 μg per 106 cells/100 μL), or CD133 (1:300). Cells were washed three times with PBS and incubated with the appropriate secondary antibody (1 μg per 10^6^ cells) for 1 h at room temperature. Cells were washed and analyzed on a BD FACSCAN flow cytometer. (**G**) Comparison of the number of positive cells for MCF-7, ZR-75-1, and MDA-MB-231 PA and SP. Data from FACS analysis assay representative of stem cell marker expression was quantified for comparison between PA and SP for all three cell lines ± SEM of three independent experiments performed in triplicates by unpaired two-tailed *t*-test at 95% confidence. **p* ≤ 0.05, ****p* ≤ 0.001, *n* = 3. Abbreviations: PA, parental; SP, spheroid; PBS, phosphate-buffered saline; CD, cluster of differentiation.

**Figure 6 cancers-13-02784-f006:**
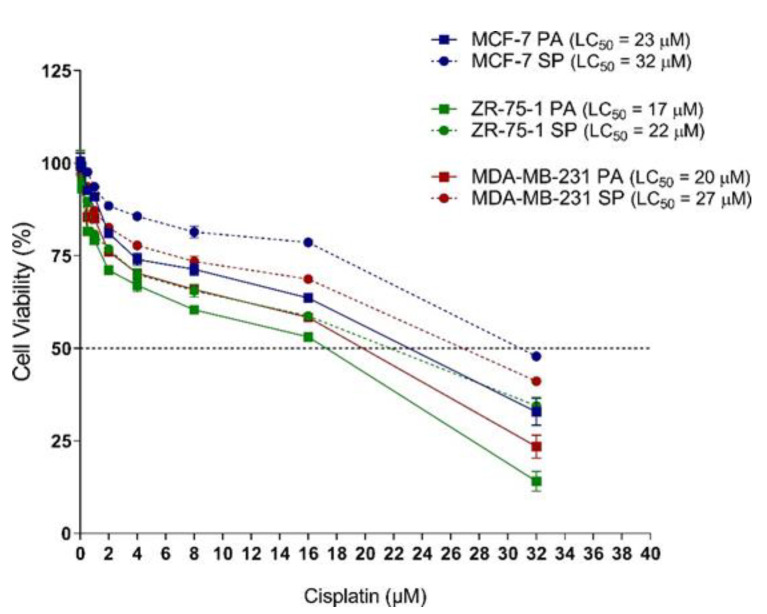
3rd generation spheroids have increased chemoresistance to cisplatin compared to monolayers. MCF-7, ZR-75-1, and MDA-MB-231 2D monolayer cells (PA) and 3rd generation spheroids (SP) were treated with 0–40 µM cisplatin for 72 h and assessed for drug resistance using the AlamarBlue cytotoxicity assay. The AlamarBlue reagent was added as 10% of the total volume to each well for 4 h before fluorometric detection. Fluorescence was measured using the SPECTRAmax Gemini Spectrophotometer (λex/λem: 540 nm/590 nm). LC50 values from the AlamarBlue cytotoxicity assay were quantified for comparison between PA and SP for all three cell lines and are representative of one out three independent experiments performed in triplicates. Abbreviations: PA, parental; SP, spheroid. LC50, the cytotoxic concentration that kills 50% of a test sample.

**Figure 7 cancers-13-02784-f007:**
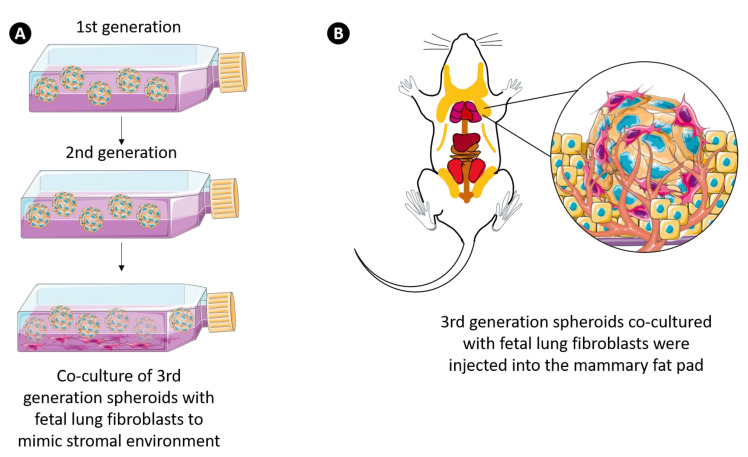
Inoculation of co-culture of spheroids and fetal lung fibroblasts into the mammary fat pad of nude rats. (**A**) 3rd generation breast cancer cell spheroids admixed with fetal lung fibroblast cells to mimic the stromal environment. (**B**) 3rd generation spheroids co-cultured with FLF were orthotopically transplanted into the mammary fat pad of nude rats. Abbreviations: FLF, fetal lung fibroblast; SP, spheroid.

**Figure 8 cancers-13-02784-f008:**
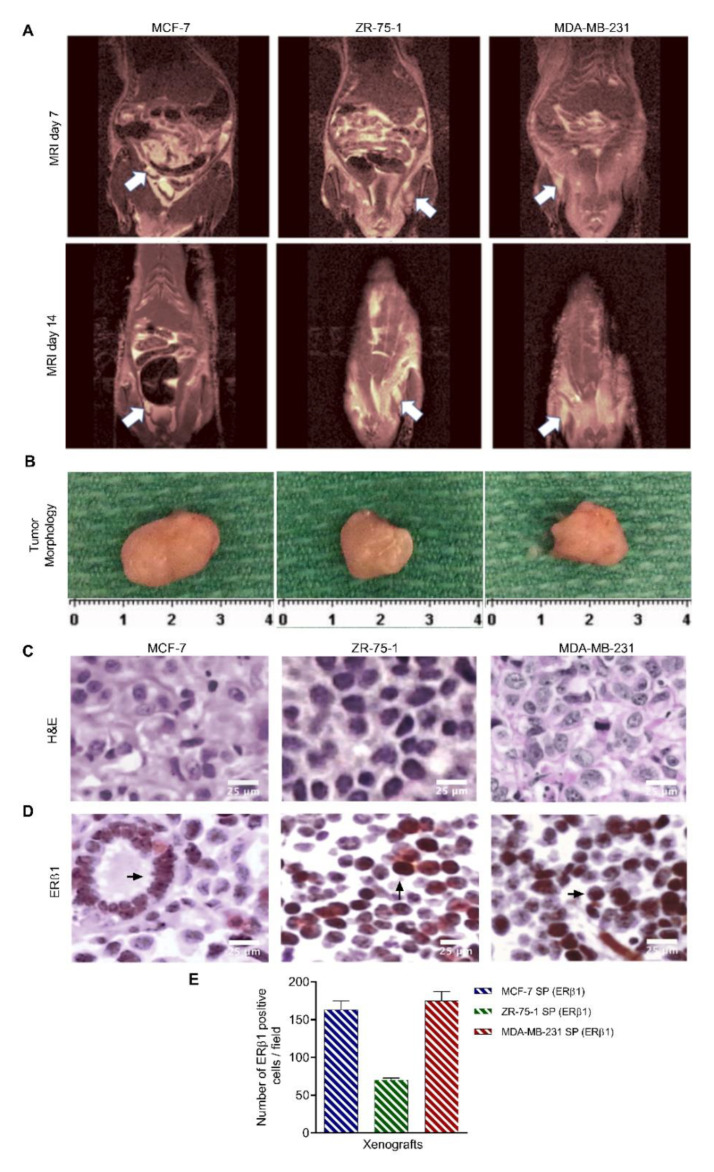
3rd generation spheroids from MCF-7, ZR-75-1, and MDA-MB-231 cells co-cultured with fetal lung fibroblast cells implanted in the fourth inguinal mammary fat pad result in tumor formation. 3rd generation spheroids admixed with FLF cells for 24 h were inoculated as 5 × 10^6^ viable cells mixed with Matrigel. Cells were injected into the fourth mammary fat pad of nude rates (*n* = 6). Rats in the control group (*n* = 9) received injections of 2D monolayer cells co-cultured with FLF mixed with Matrigel. (**A**) MRI of tumor mass. Tumors are detected as a hyperintense signal on T2-weighted images. (**B**) Tumors were excised before they reached the permitted endpoint (1.5 cm in diameter). (**C**) H&E histology of resected tumors fixed in 10% neutral buffered formalin at room temperature and embedded in paraffin, sectioned into 5 μm thick slices. (**D**) Paraffin sections of resected tumor tissues were stained with primary antibodies against the estrogen receptor β1 (ERβ1) (1:200). Immunoperoxidase labeling was performed using the ABC Peroxidase Standard Staining Kit. Images of stained tumor tissue were taken using a light microscope. (**E**) Data from immunoperoxidase labeling assay. Bar graph representative of the number of positive ERβ1 cells per field ±SEM, was quantified for xenografts grafted from SP for all three cell lines ±SEM, *n* = 5. Abbreviations: PA, parental; SP, spheroid; ERβ1, estrogen receptor β1; 2D, 2-dimensional.

**Figure 9 cancers-13-02784-f009:**
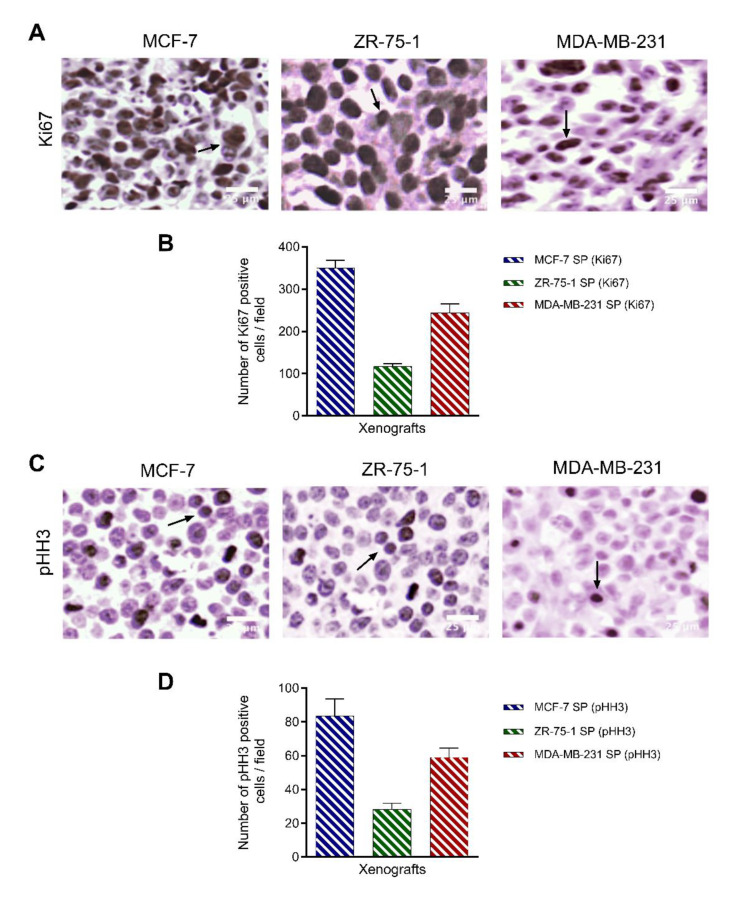
3D stem-like spheroid-derived breast tumor xenografts express proliferation Ki67, and mitotic pHH3 markers. Parafilm sections (5 µm) from xenografts generated from MCF-7, ZR-75-1, and MDA-MB-231 spheroids were stained for Ki67 and pHH3. Images were taken using a light microscope. (**A**) Tissue was stained with primary antibody against proliferation marker Ki67 (1:250). (**B**) Graphs of the number of positive Ki67 immunostained cells per field ±SEM; experiments performed in triplicates. (**C**) Tissue was stained with primary antibody against proliferation marker pHH3 (1:250). (**D**) Graphs of the number of positive pHH3 immunostained cells per field ±SEM; experiments performed in triplicates.

**Figure 10 cancers-13-02784-f010:**
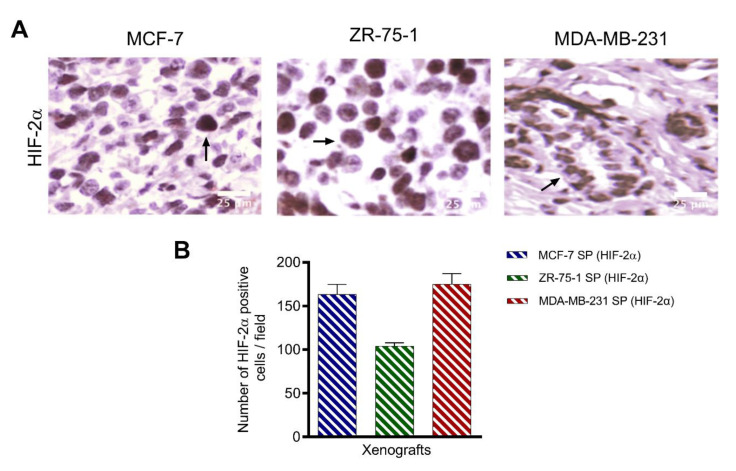
Expression of hypoxia marker HIF-2α on 3D stem-like derived spheroid breast tumor xenografts. Parafilm sections (5 µm) from xenografts generated from MCF-7, ZR-75-1, and MDA-MB-231 spheroids were stained using a primary antibody against HIF-2α (1:200). (**A**) Images were taken using a light microscope. (**B**) Graph of the number of positive HIF-2α immunopositive cells per field ±SEM; three independent experiments performed in triplicates.

**Figure 11 cancers-13-02784-f011:**
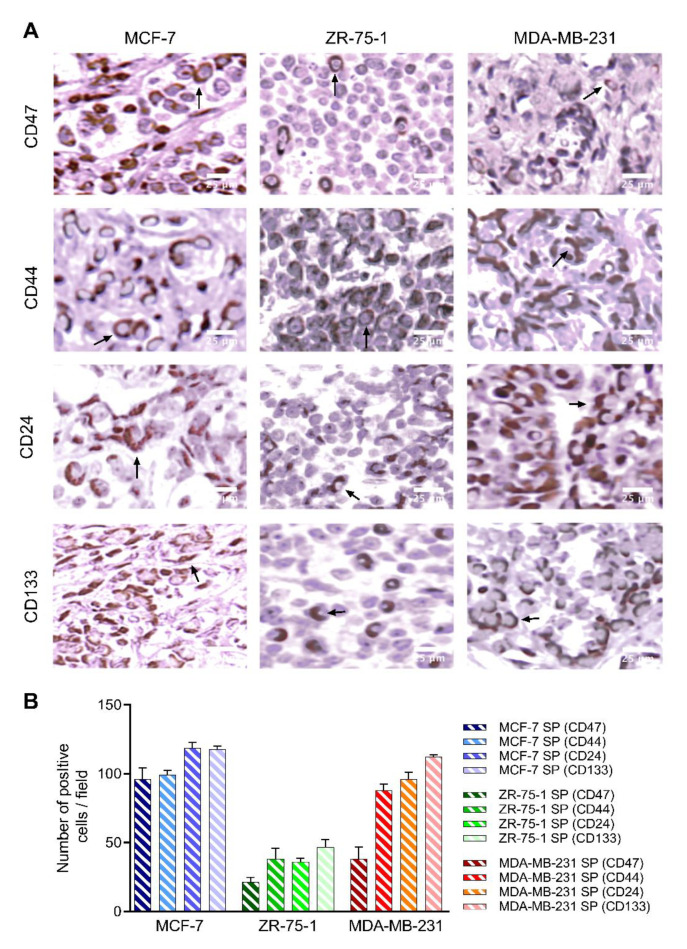
Expression of CD47, CD44, CD24, and CD133 stem cell markers on 3D stem-like derived spheroid breast tumor xenografts. Parafilm sections (5 µm) from xenografts generated from MCF-7, ZR- 75-1, and MDA-MB-231 spheroids were stained using a primary antibody against CD47 (1:50), CD44 (1:50), CD24 (1:50), or CD133 (1:200). (**A**) Images were taken using a light microscope. (**B**) Graph of the number of positive stem cell marker immunostained cells per field ±SEM; three independent experiments performed in triplicates.

**Figure 12 cancers-13-02784-f012:**
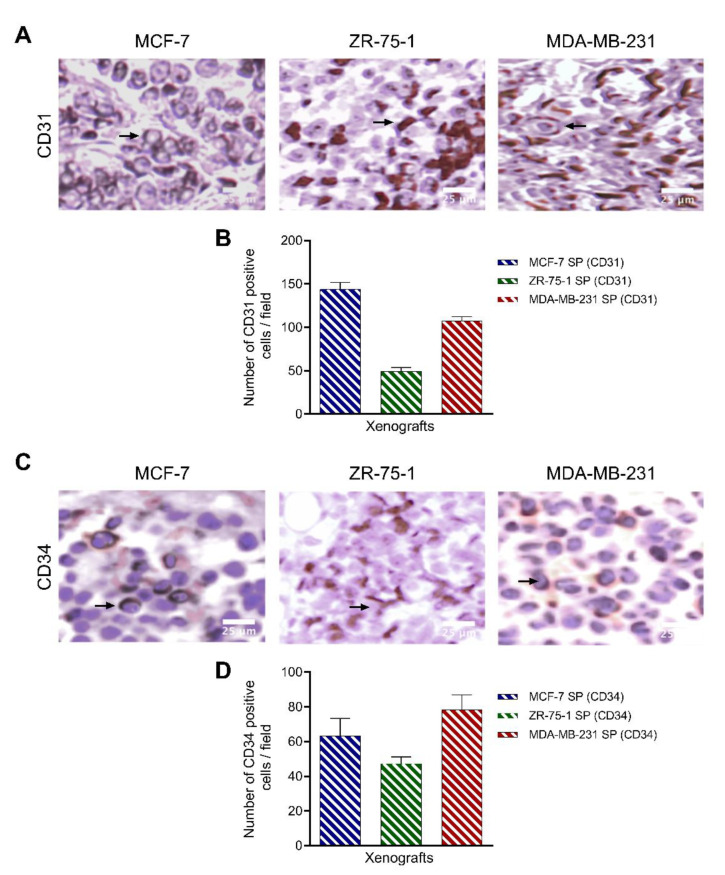
3D stem-like derived spheroid breast tumor xenografts express vascularization markers. Light microscope images of H&E-stained tumor tissue segments from xenografts generated from MCF-7, ZR-75-1, and MDA-MB-231 spheroids were stained using primary antibodies against (**A**,**B**) CD31 (1:50) or (**C**,**D**) CD34 (1:2500) to determine the degree of vascularization. Graphs show the number of positive vascularization marker immunostained cells per field ±SEM; three independent experiments performed in triplicates.

**Figure 13 cancers-13-02784-f013:**
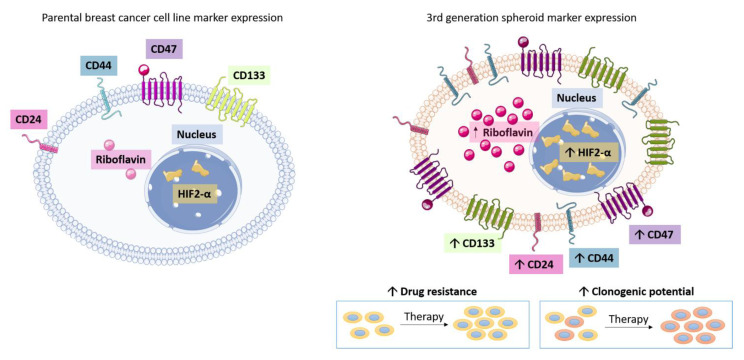
Characterization and comparison of 2D monolayer breast cancer cells and 3rd generation spheroids. 3rd generation breast cancer spheroids express increased CSC markers CD133, CD24, CD44, and CD47 expression; elevated riboflavin accumulation; and increased HIF-2α, compared to the monolayer counterparts. Abbreviations: CSC, cancer stem cell.

## Data Availability

The data is presented in this study.

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
