# Peer review of "3D Multicellular Stem-Like Human Breast Tumor Spheroids Enhance Tumorigenicity of Orthotopic Xenografts in Athymic Nude Rat Model"

_cancers, 2021, doi:10.3390/cancers13112784_

Round 1

Reviewer 1 Report

In this manuscript,  Mokhtari RB et al., studied 3D multicellular stem-like human breast tumor spheroids enhance tumorigenicity of orthotopic xenografts in athymic nude rat model . In this study, developing 3D multicellular stem-like properties of breast cancer cells under stem cell-enriched medium and spheroid-promoting conditions were evaluated using three different breast cancer cell lines, MCF-7, ZR-75-1, and MDAMB-231 and these cells were formed spheroids which exhibiting increased stemness characteristics when cultured in stem cell factor supplement media. When the spheroids were co-cultured with human FLFsas a stromal component, they yielded a reproducible and high take rate efficiency in establishing orthotopic xenografts exhibiting cancer stem cell phenotype, hypoxia, and increased vascularization in the athymic nude rat model of human breast cancer and this approach author trying to develop novel breast cancer diagnostic and therapeutic model.  I would like to recommend Editor to accept this manuscript as such without any changes in this journal.

Author Response

I would like to recommend Editor to accept this manuscript as such without any changes in this journal.

Thank you for your positive review of the manuscript.

Reviewer 2 Report

The authors present a very detailed study into the stem cell characteristics of 3D spheroids vs. 2D cultures, and their behaviour after xenotransplantation. The characterization of the 3D spheroids is done to a great extent, using a variety of markers, which correlate very well.

However, it is unclear to me what the primary claim of this manuscript it. Are the authors claiming that 3D spheroids co-cultured with FLFs better recapitulate the TME, and lead to higher rate of tumor formation? In this case there is no direct comparison with 3D spheroids frown without FLFs. The comparison in marker expression has been done between 3D spheroids, and co-cultured 3D spheroids after implantation. The authors need to add the comparison to in vitro co-cultured spheroid, and spheroid cells alone after implantation.

I also find it strange that not a single mouse formed a tumor after implantation of 5 million cells (2D culture). The rate of tumor formation in mice is much higher, and if it is this low in rats, appropriate references should be provided. Also, they should try injecting more cells. It is hard to compare efficacies, if the efficacy of one technique is zero.

In addition to these controls, the authors should rewrite the manuscript so that the claim is clear. Is it that 3D spheroids with FLFs recapitulate effectf of TME better? or is the primary claim that 3D spheroids are an effective method for xenotransplantation in rats.

Author Response

Comments and Suggestions for Authors:

Reviewer comment: The authors present a very detailed study into the stem cell characteristics of 3D spheroids vs. 2D cultures, and their behaviour after xenotransplantation. The characterization of the 3D spheroids is done to a great extent, using a variety of markers, which correlate very well.

However, it is unclear to me what the primary claim of this manuscript it. Are the authors claiming that 3D spheroids co-cultured with FLFs better recapitulate the TME, and lead to higher rate of tumor formation? In this case there is no direct comparison with 3D spheroids frown without FLFs. The comparison in marker expression has been done between 3D spheroids, and co-cultured 3D spheroids after implantation. The authors need to add the comparison to in vitro co-cultured spheroid, and spheroid cells alone after implantation.

Author response: Thank you for the comment. In the revised manuscript, the journal requested a simple summary of the study. Here, we have indicated the aim of the study. For example, “The aim of this study was to investigate breast cancer cells cultured in stromal stem cell factor-supplemented media to generate 3D spheroids that exhibit increased stem-like properties. These 3D stem-like spheroids reproducibly and efficiently established orthotopic breast cancer xenografts in the athymic nude rat. This approach enables a means to develop orthotopic tumors with a stem-like phenotype in larger athymic rat rodent model of human breast cancer.

The question concerning Fetal Lung fibroblast cells. FLF cells were co-cultured with 3D spheroids breast cancer cells to better recapitulated the TME and potentially act as a cancer-associated fibroblasts (CAFs) elements. Previous studies were shown that they are a key component of the tumor microenvironment (TME) with diverse functions, including matrix deposition and remodeling, extensive reciprocal signaling interactions with cancer cells, and crosstalk with infiltrating leukocytes (Sahai, E. et al. A framework for advancing our understanding of cancer- associated fibroblasts. Nature Reviews, Cancer, https://www.nature.com/articles/s41568-019-0238-1.pdf). While it would have been informative to compare against injecting 3D spheroids without FLFs, that was not the primary purpose of this paper. Our goal was to recapitulate the TME as best as we could, which entails both 3D geometry and FLFs, to attain our singular objective of improving the take-rate from zero percent success.

As mentioned in the discussion (at end of the first paragraph), in our previous study [ref 23], we injected 3D spheroids of bronchial carcinoid with and without FLF in NOD/SCID mice. There was a greater take rate in the spheroids admixed with FLF compared to the spheroids alone. Conversely, when we repeated experiments with breast cancer cells in NOD/SCID mice (not humanized mice), we saw no success in the development of xenografts without FLF as exogenous hormones were proposed necessary due to the hormone-dependent and hormone-independent nature of breast cancer cell lines. Based on our previous experiments on the NOD/SCID mouse model, we decided to develop xenografts using breast cancer cells co-cultured with FLF injected into the nude rat.

Reviewer comment: I also find it strange that not a single mouse formed a tumor after implantation of 5 million cells (2D culture). The rate of tumor formation in mice is much higher, and if it is this low in rats, appropriate references should be provided. Also, they should try injecting more cells. It is hard to compare efficacies, if the efficacy of one technique is zero

Author response: The low uptake in rats is a key reason why immune-compromised mice are used for cancer studies. It is possible for rats to take up tumors, but this is seen for very aggressive metastatic tumors, such as the LM2 model that we worked on previously (Nofiele and Cheng. Establishment of a lung metastatic breast tumor xenograft model in nude rats. PLoS One 2014; 9(5):e97950). In collaboration with other investigators at other institutions, we have first-hand knowledge that it is universally very difficult for rats to take up tumors, which may be due to the “leaky” immune system of nude rats.

To respond to the second concern, we firstly needed to correct two important information in Reviewer #2 statements: a) we inoculated both breast cancer cells (3D spheroid and 2D monolayer) in the athymic nude rat model and not in the mouse model; b) secondly, a 1:3 ratio of FLF(1.6 million) and 3rd generation spheroids (3.4 million) co-cultured or 1:3 ratio of FLF (1.6 million) and 2D monolayer (3.4 million) co-cultured inoculated to nude rat; however, Reviewer #2 stated that 5 million breast cancer cells.

Data on take rates in rats is scarce, most likely because rats present an even greater challenge. To fully utilize the rat as a model for breast cancer studies, we require a highly reproducible method for successfully establishing breast tumor xenografts in rats

We have added the following in the discussion (2nd paragraph) of the revised manuscript.Xenografting typically involves implanting tumor fragments or tumor cells that have been cultured in a traditional two-dimensional (2D) monolayer environment. With this approach, a recent study showed that the take rate varied from 0% when human MDA-MB-231 breast cancer cells were injected subcutaneously in nude rats to 25% when matrigel was co-injected [Direcks WG, van Gelder M, Lammertsma AA, Molthoff CF (2008) A new rat model of human breast cancer for evaluating efficacy of new anti-cancer agents in vivo. Cancer Biol Ther 7: 532-537.7]. An alternative to 2D monolayer culturing is the three-dimensional (3D) spheroid culture method. Multicellular spheroids are important in-vitro models as they can mimic the functional and architectural characteristics of in-vivo tissues [Hirschhaeuser F, Menne H, Dittfeld C, West J, Mueller-Klieser W, et al. (2010) Multicellular tumor spheroids: an underestimated tool is catching up again. J Biotechnol 148: 3-15]. For example, they enable complex cell-to-cell interactions and establish gradients (e.g. nutrients, oxygen, metabolites) as a result of barriers to diffusion imposed by the spheroid. In cancer biology, spheroids are an important model for drug discovery and are commonly used in the in-vitro setting to understand cell-stromal interactions and mechanisms of cancer metastasis [Achilli TM, Meyer J, Morgan JR (2012) Advances in the formation, use and understanding of multi-cellular spheroids. Expert Opin Biol Ther 12: 1347-1360.]. To use spheroids directly in vivo it is a new approach. Such reports are rare, but a recent study reported an interesting finding that mouse tumor xenografts from spheroid injections were larger than those from adherent cells [Mu Z, Li H, Fernandez SV, Alpaugh KR, Zhang R, et al. (2013) EZH2 knockdown suppresses the growth and invasion of human inflammatory breast cancer cells. J Exp Clin Cancer Res 32: 70]. The idea that 3D spheroids may improve establishment of tumo.

Reviewer comment:In addition to these controls, the authors should rewrite the manuscript so that the claim is clear. Is it that 3D spheroids with FLFs recapitulate effectf of TME better? or is the primary claim that 3D spheroids are an effective method for xenotransplantation in rats?

Author response: The claim of this manuscript is recapitulating the true TME through the combined use of 3D spheroids and FLFs enables the successful uptake of a variety of breast tumors in nude rats. We have edited the manuscript to clarify this claim.

In this study, we sought to develop a reproducible method for inducing breast tumor xenografts in nude rats. It was shown that the 3D spheroid culture method successfully established different human breast cancers (ZR-75-1, MCF-7, and MDA-MB-231) orthotopically, with a take rate of 100%, whereas conventional 2D monolayer culturing failed to establish tumors for any of the breast cancer subtypes. All tumors displayed typical hyperintensity on T2-weighted MRI. Histology confirmed apoptosis and active proliferation in all tumors. The degree of vascularity and hypoxia varied with tumor subtype, but generally low vascularity was observed.

A success rate of 100% for establishing orthotopic breast tumors is exceptional, considering that breast cancer is one of the more difficult tumors to establish in experimental animals, including nude mice and SCID mice, and considering that take rates are even lower in rats.

Therefore, this study demonstrated that 3D stem-like spheroids co -culture of human adenocarcinomas (ZR-75-1, MCF-7, and MDA-MB-231) with fetal lung fibroblast cells is highly efficient for establishing orthotopic human breast tumors in nude rats. The success was 100% for all three cancer subtypes, whereas conventional 2D monolayer culture of cancer cells failed to induce tumors in vivo. Monitoring by MRI revealed tumors were seen by Day 7 and 14 after cell inoculation. Histology confirmed hypoxic and proliferative tumors with low vascularity. The 3D spheroid culture method is valuable for establishing the nude rat as a viable model for studying tumor progression, metastasis, and treatment in larger animals.

Reviewer 3 Report

In this paper, Mokhtari and coworkers developed a xenograph rat model of breast cancer that confirms the importance of stromal cells in regulating stemness. The state-of-the-art is adequate to the field of research. Methodology is well described and also adequate. However, the paper does not provide new knowledge about the role of fibroblasts in the evolution of cancer stem cells towards phenotypes more resistant to standard treatments. In addition, following points should be addressed:

  1. In the paper, a group of animals inoculated with spheroids not co-cultured with lung fibroblasts is missed.
  2. Since the strength of the paper is the development of the xenograph model, the demonstration of the role of fibroblasts in the development of resistance to therapy in breast cancer, the authors should also include two groups of treatment with cisplatin: 1) animals inoculated with co-cultured speroids with fibroblasts, and 2) animals inoculated with esperoids not co-cultured with fibroblasts.
  3. Abstract must have a maximum length of 200 words
  4. There are a lot of typo in the text. Please correct them.

Author Response

Reviewer # 3: Comments and Suggestions for Authors

In this paper, Mokhtari and coworkers developed a xenograph rat model of breast cancer that confirms the importance of stromal cells in regulating stemness. The state-of-the-art is adequate to the field of research. Methodology is well described and also adequate. However, the paper does not provide new knowledge about the role of fibroblasts in the evolution of cancer stem cells towards phenotypes more resistant to standard treatments. In addition, following points should be addressed:

1.In the paper, a group of animals inoculated with spheroids not co-cultured with lung fibroblasts is missed.

Author response: This was not the objective of the study. Our goal was to achieve successful tumor uptake in nude rats, and we approached this goal by replicating both the 3D stem-like spheroids in the presence of FLFs. As discussed above to Reviewer #2 comment, the claim of this manuscript is recapitulating the true TME through the combined use of 3D stem-like spheroids and FLFs which enables the successful uptake of a variety of breast tumors in nude rats. We have edited the manuscript to clarify this claim.

2.Since the strength of the paper is the development of the xenograph model, the demonstration of the role of fibroblasts in the development of resistance to therapy in breast cancer, the authors should also include two groups of treatment with cisplatin: 1)animals inoculated with co-cultured speroids with fibroblasts, and 2) animals inoculated with esperoids not co-cultured with fibroblasts.

Author response: The goal was not to investigate the differential efficacy achieved with FLFs.Our focus of study in the present study was not drug resistance. The only reason that we treated 3D stem-like spheroids with cisplatin (in-vitro) was to show that our 3D stem-like spheroids have drug resistance capacity which is one of the important criteria of stem cell characterization. Abstract must have a maximum length of 200 wordsAuthor response:The revised manuscript has an abstract of less than 200 words. We have included a Simple Summary where we indicated the aim of the study. For example, “The aim of this study was to investigate breast cancer cells cultured in stromal stem cell factor-supplemented media to generate 3D spheroids that exhibit increased stem-like properties. These 3D stem-like spheroids reproducibly and efficiently established orthotopic breast cancer xenografts in the athymic nude rat. This approach enables a means to develop orthotopic tumors with a stem-like phenotype in larger athymic rat rodent model of human breast cancer.”

3.There are a lot of typo in the text. Please correct them.

Author response:We have corrected typographical errors.

Round 2

Reviewer 2 Report

The authors have satisfactorily clarified all my concerns.

Reviewer 3 Report

I think the paper is ready for publication in the journal